# Antidepressant Drugs and COVID-19: A Review of Basic and Clinical Evidence

**DOI:** 10.3390/jcm11144038

**Published:** 2022-07-12

**Authors:** Marta Mas, Juan Antonio García-Vicente, Anaïs Estrada-Gelonch, Clara Pérez-Mañá, Esther Papaseit, Marta Torrens, Magí Farré

**Affiliations:** 1Medical Writing Department, TFS HealthScience, 08007 Barcelona, Spain; marta.mas@tfscro.com (M.M.); anais.estrada@tfscro.com (A.E.-G.); 2Experimental and Health Sciences Department, Universitat Pompeu Fabra (UPF), 08002 Barcelona, Spain; 3Department of Pharmacology, Therapeutics and Toxicology, Universitat Autònoma de Barcelona (UAB), Cerdanyola del Vallès, 08290 Barcelona, Spain; cperezm.mn.ics@gencat.cat (C.P.-M.); epapaseit.germanstrias@gencat.cat (E.P.); mtorrens@parcdesalutmar.cat (M.T.); 4Pharmacy Unit, Northern Metropolitan Primary Care Directorate, Catalan Institute of Health, 08911 Barcelona, Spain; 5Clinical Pharmacology Unit, Hospital Universitari Germans Trias i Pujol and Institut de Recerca Germans Trias i Pujol (HUGTP-IGTP), 08916 Barcelona, Spain; 6Addiction Unit, Hospital del Mar Medical Research Institute (IMIM), Institut de Neuropsiquiatria i Addiccions, Parc de Salut Mar, 08003 Barcelona, Spain

**Keywords:** COVID-19, SARS-CoV-2, antidepressant, cytokine storm, antiviral, fluoxetine, fluvoxamine

## Abstract

The COVID-19 pandemic has encouraged the repurposing of existing drugs as a shorter development strategy in order to support clinicians with this difficult therapeutic dilemma. There is evidence to support the theory that some antidepressants can reduce concentrations of different cytokines in humans and animals and, recently, the antiviral activity of some antidepressants against SARS-CoV-2 has been reported. The aims of this narrative review are to evaluate the possible role of antidepressants in the treatment of COVID-19 infection and the possible benefits and risks of patients taking antidepressants for mental disorders and COVID-19 infection. A review was performed to analyse the current literature to identify the role of antidepressant medication in the treatment of COVID-19 patients. The electronic search was completed in MEDLINE and MedRxiv/BioRxiv for published literature and in ClinicalTrials.gov for ongoing clinical trials. The results show some evidence from preclinical data and observational studies about the possible efficacy of some specific antidepressants for treating COVID-19 infection. In addition, two published phase II studies testing fluvoxamine showed positive results for clinical deterioration and hospitalization rate versus a placebo. Seven ongoing clinical trials testing fluvoxamine, fluoxetine, and tramadol (as per its anti-inflammatory and antidepressant effect) are still in the early phases. Although the available evidence is limited, the sum of the antiviral and anti-inflammatory preclinical studies and the results from several observational studies and two phase II clinical trials provide the basis for ongoing clinical trials evaluating the possible use of antidepressants for COVID-19 infection in humans. Further investigations will be needed to support the possible use of antidepressants for this application.

## 1. Introduction

Coronaviruses are potential infectious agents transmitted between species that are highly prevalent and show great genetic diversity [1,2]. Of the six coronavirus species that cause disease in humans, four of them (229E, OC43, NL63, and HKU1) cause common cold symptoms in immunocompetent people [2]. The other two strains are the coronaviruses that cause severe acute respiratory syndrome (SARS-CoV) and Middle East respiratory syndrome (MERS-CoV), which can cause fatal respiratory illnesses in humans [3,4,5,6]. SARS-CoV-2 was identified as a new betacoronavirus causing the disease called COVID-19 [7], which was declared a pandemic in March 2020 [8]. As of 25 June 2022, more than 543 million cases of COVID-19 have been confirmed worldwide, with more than 6,328,000 deaths [9]. The transmission of SARS-CoV-2 occurs through respiratory droplets, and its spread depends on population density, the prevalence of underlying medical conditions, the ability to perform diagnostic tests, etc. [10]. Factors associated with disease severity have been described and include age over 60 years, male sex, obesity, hypertension, diabetes, and chronic obstructive pulmonary disease [11,12].

The causative virus (SARS-CoV-2) of COVID-19 has evolved over time and different variants (variants of concern) have appeared, replacing the previous ones, each with their own peculiarities in transmission, evasion of immunity, and symptomatology. Each of these variants of concern have been designated a letter of the Greek alphabet (alpha, beta, gamma, delta, and omicron).

When the pandemic began, amino acid changes in the SARS-CoV-2 spike protein were monitored. One strain with a substitution of D614G (glycine for aspartic acid) was identified and became the dominant polymorphism worldwide [13]. The G614 variant is present in most circulating SARS-CoV-2 lineages, including the delta variant of concern (lineage B.1.617.2), which was first identified in India in December 2020. After that, it became the most prevalent variant worldwide until the appearance of the Omicron variant (B.1.1.529). This variant was first detected in Botswana, and shortly thereafter in South Africa in November 2021. In South Africa, it was associated with an increase in infections and spread rapidly through many countries [14,15]. As of the end of December 2021, Omicron accounted for most new infections in the United States. This variant contained more than 30 mutations in the spike protein; these are associated with increased transmissibility [16,17] and allow the variant to evade infection- and vaccine-induced humoral immunity to a greater extent than previous variants [18,19]. According to the results of observational studies, it is now considered that Omicron may lead to a less severe clinical picture than other variants [20,21].

COVID-19 disease includes a wide range of clinical manifestations, ranging from asymptomatic presentations or mild infections (81%: non pneumonia and mild pneumonia) to severe viral pneumonia (14%) and critically ill patients (5%: respiratory failure, septic shock, and/or multiple organ failure or dysfunction) [22] (Wu & McGoogan, 2020). The most common symptoms are fever, cough, and dyspnoea [12]. Diarrhoea is uncommon. The median incubation period is four days, and the median time until the development of pneumonia is five days after the onset of COVID-19 symptoms [12].

Ground-glass opacity is the most common radiologic finding in chest computed tomography (CT), and many scans do not lead to abnormal radiologic findings (no radiographic or CT abnormality was found in 18% of patients with non-severe disease and in 3% with severe disease) [12]. On admission, lymphopenia is the most frequent hematologic disorder, followed by thrombocytopenia and leukopenia. Most patients have elevated levels of C-reactive protein and, less commonly, elevated levels of D-dimer. Patients with severe disease have more significant laboratory abnormal parameters than those with non-severe disease [12].

Numerous studies that aim to determine the virological [23,24], epidemic [25,26], clinical [27], and pathological [28] characteristics of COVID-19 have been published, as well as studies on prognostic predictive clinical and analytical factors [29,30]. SARS-CoV-2 causes more severe clinical conditions in patients aged ≥60 years and with coexistent medical comorbidities. Regarding gender differences, male patients present a higher risk of developing severe illness and increased rates of mortality [31].

### 1.1. Pathophysiology of COVID-19

To understand the pathophysiology of the disease and use the appropriate treatments, different stages of the disease must first be distinguished. In the first phase, viral pathogenicity predominates, and in the second, a state of systemic inflammatory by the host response is enhanced [30].

#### 1.1.1. Viral Phase

SARS-CoV-2 infects different human cells by binding to the cell surface protein angiotensin-converting enzyme 2 (ACE2) through the receptor-binding domain (RBD) of its spike (S) protein in the respiratory epithelium. ACE2 receptors are also expressed by other organs, such as the oesophagus, ileum, myocardial cells, proximal tubular cells of the kidney, and bladder urothelial cells. Cellular transmembrane serine protease 2 (TMPRSS2) is required for the priming of the virus S protein, which facilitates cell entry and viral replication [30].

Sphingosine and ceramide are sphingolipids that are found in cell membranes. Acid sphingomyelinase transforms sphingosine to ceramide in lysosomes and cell membranes. These lipids are known to be involved in the development of bacterial and viral infections. Specifically, they have been shown to interfere with the uptake of SARS-CoV-2 viral particles in cultured human nasal cells and epithelial cell lines [32]. SARS-CoV-2 infection activates the acid sphingomyelinase system and triggers the release of ceramide on the cell surface, which favours SARS-CoV-2 infection [33,34], while sphingosine blocks the entry of the virus [32]. Data indicating that the serum levels of sphingosine-1-phosphate (S1P) could be a prognostic factor for the severity of COVID-19 are available.

Although SARS-CoV-2 has a high tropism for ACE2-expressing respiratory epithelial cells, people with severe COVID-19 infections have symptoms of systemic hyperinflammation.

#### 1.1.2. Inflammatory Phase

This phase is the most severe form of the disease and will occur in a minority of COVID-19 patients. At this stage, patients with more severe disease have a significant elevation of cytokines; biomarkers such as IL-2, IL-6, and IL-7; granulocyte colony-stimulating factor; tumour necrosis factor-α (TNF-α); macrophage inflammatory protein 1-a; D-dimer; C-reactive protein; and ferritin. Virus particles spread through the respiratory mucosa and infect other cells, inducing a cytokine storm in the body and generating a series of immune responses [29].

Cytokine storm syndrome (CSS) is a massive release of cytokines that occurs in a wide variety of infectious and non-infectious diseases [35]. This heightened immune response causes multi-organ dysfunction and can even lead to death [36]. Although the initial drivers of such a response may differ, in the later stages, the clinical manifestations tend to converge and overlap [37]. In the early 2000s, CSS associated with infections caused by bacteria and viruses was described, and more recently these infections have included severe acute respiratory syndrome coronavirus (SARS-CoV) [38].

SARS-CoV-2 infection provokes innate and adaptive immune responses that may contribute to the severity of the disease [39] and be the underlying cause of the most severe clinical manifestations, which are associated with the overproduction of proinflammatory cytokines [40] and lead to increased vascular hyperpermeability and multi-organ failure [41]. During the progression of COVID-19, the hyperinflammatory response is related to high levels of cytokines, IL-β, IL-2, IL-6, IL-7, IL-10, GCSF, IP-10, MCP-1, MIP-1A, and TNF-α, and VEGF [37,42,43,44]. In addition, a general immune dysregulation is observed, with total white blood cell and neutrophil counts being significantly higher in the most severe cases, while lymphocyte counts decrease. The overproduction of proinflammatory cytokines is mediated by monocytes [45]. An increase in T cell apoptosis could be a possible explanation for the decrease in CD4+ and CD8+ lymphocytes observed, which may be the result of high serum concentrations of TNF-α, IL-6, and IL-10 [46]. Moreover, markers of depletion, such as NKG2A receptors on NK cells and CD8+ T cells, are upregulated in COVID-19 patients, while CD4+ T and CD8+ T cells are downregulated in lymph nodes and the spleen [47]. This behaviour of T cells, known as exhaustion or senescence, has been observed in patients with COVID-19 pneumonia [48]. All this together explains why the impact of SARS-CoV-2 does not correlate with a direct effect of the virus on cells, but rather with the hyperinflammatory response.

The possibility of repurposing antidepressants for the treatment of COVID-19 comes from findings regarding antiviral screening activity; epidemiological retrospective data from patients; recent non-controlled series of cases; and two randomized placebo-controlled clinical trials evaluating the use of fluvoxamine in non-hospitalized patients, which showed promising positive results.

The aims of this narrative literature review are to evaluate the possible role of antidepressants in the treatment of coronavirus infections and identify the possible benefits and risks of patients taking antidepressants for mental disorders and coronavirus infection. 

This literature review was based on an electronic search completed on MEDLINE and MedRxiv/BioRxiv for published literature and on ClinicalTrials.gov, the EU Clinical Trials Register, and Chinese Clinical Trial Registry (https://www.chictr.org.cn/abouten.aspx, accessed 10 May 2022) for ongoing clinical trials.

## 2. Non-Clinical Research Evidence

### 2.1. Antiviral Activity of Antidepressants

Recent evidence shows that some antidepressants can have direct antiviral effects. A recent in vitro study suggests that fluoxetine has antiviral effects on SARS-CoV-2, but these effects were not observed for other selective serotonin reuptake inhibitors, including paroxetine and escitalopram [49]. Fluoxetine inhibited SARS-CoV-2 at a concentration of 0.8 μg/mL, and the EC50 was determined as 387 ng/mL. Fluoxetine isomers showed similar activity against the virus. The mechanism of action was unknown, but fluoxetine produced a decrease in viral protein expression. In another study, fluoxetine was shown to inhibit SARS-CoV-2 infection in a dose-dependent relationship, with an EC50 value of below 1 μM. The application of 10 μM of fluoxetine reduced viral titers by up to 99%, suggesting a mechanism related to the functional inhibition of sphingomyelinase and acting on the endolysosomal acidification [50]. In another study, fluoxetine showed antiviral activity and was demonstrated to inhibit SARS-CoV-2 in vitro [51]. The combination of remdesivir and fluoxetine showed antiviral activity inhibiting the production of infectious SARS-CoV-2 particles by more than 90% in a polarized Calu-3 cell culture model. Similar results were observed after the use of a combination of itraconazole and remdesivir [52].

In one study, the anti-SARS-CoV-2 potential of 1403 FDA-approved drugs was quantitatively studied by a pseudovirus-based assay [53]. Vortioxetine, an atypical antidepressant known as a serotonin modulator and stimulator, showed antiviral action at different concentrations. The IC50 and IC90 values were 3 μmol and 13.8 μmol, respectively.

The cytopathic effect of SARS-CoV-2 can be blocked by autophagy modulators [54]. A series of different compounds have been screened in order to evaluate the potential antiviral effects of this mechanism. The autophagy inhibitors included ROC-325, hycanthone, clomipramine, verteporfin, chloroquine, hydroxychloroquine, and mefloquine. Clomipramine showed an IC50 for an autophagy of 13.2 ± 5.4 μmol, higher than that obtained by the other compounds.

One published study evaluated some inhibitors of two coronavirus spike proteins identified by screening a library of approved drugs with SARS-S and MERS-S pseudotyped particle entry assays [29]. Using high-throughput screening technology, the authors found three library compounds (abemaciclib, cepharanthine, and trimipramine) to be broad-spectrum inhibitors for spike-mediated entry. Trimipramine is a typical antidepressant that is not commonly used in therapeutics.

### 2.2. Anti-Inflammatory Properties of Antidepressants

It is known that some antidepressants can produce a significant reduction in overexpressed inflammatory processes in individuals affected by major depressive disorder [55,56]. 

In a meta-analysis of studies conducted on major depressive disorder, which included data derived from 45 longitudinal studies and more than 1500 depressive patients, the results showed that antidepressants in general, and selective serotonin reuptake inhibitors (SSRIs) and serotonin–norepinephrine reuptake inhibitors (SNRIs) in particular, reduced the plasma levels of several proinflammatory cytokines, including IL-6, TNF-α, and CCL-2, which could be involved in the pathogenesis of the cytokine storm observed in severe COVID-19, as previously noted [56]. In one study, it was suggested that SSRIs could more potently inhibit microglial TNF-α and NO production through cAMP signalling regulation than SNRIs [57].

In a study performed on preclinical models of inflammation and sepsis evaluating the action of different substances for the modulation of the sigma-1 receptor–IRE1 pathway, only 9% of the mice given fluvoxamine died, compared with 62% of the untreated mice. In addition, fluvoxamine showed relevant anti-inflammatory activity; significantly reduced ligand polysaccharide LPS-induced IL-6, IL-1alfa, and LL-12 p40; and decreased IL-8 production in cells [58]. It is known that SARS-CoV-2 Nsp6 protein interacts with the Sigma receptor, which is believed to participate in ER stress response [59,60].

A recent letter supports the hypothesis that serotonergic and noradrenergic drugs may have opposing effects on the COVID-19 cytokine storm. Since 5HT may be functionally opposed to NE, drugs that facilitate 5HT transmission may also dampen the cytokine storm related to COVID-19, including SSRIs, some tricyclic antidepressants (clomipramine), monoaminoxidase inhibitors (phenelzine, tranylcypromine), and the 5HT2C agonist lorcaserin [61]. 

Other evidence supports the possible role of clomipramine, a serotonin and noradrenaline reuptake inhibitor, in preventing neurological complications of SARS-CoV-2 infection due to decreases in pro-inflammatory cytokines. It has been demonstrated to have consistent anti-inflammatory properties at therapeutic concentrations, as shown in many studies (in vitro, in vivo, and in humans) [62]. The possible immunoregulatory and anti-inflammatory effects of SSRIs in COVID-19 patients have been reviewed. SSRIs could play a role in COVID-19 infection through treating stress and anxiety, increasing the number and function of immune cells, and reducing cytokine release syndrome by reducing IL-6 and IL-10 levels [63].

Tryptophan metabolism and kynurenines have been found to be related to inflammation and immunity. In one study analysing the metabolome profile of 55 patients infected with SARS-CoV-2, the role of the tryptophan nicotinamide pathway seems to be clearly linked to inflammatory signals and microbiota, as well as to the possible involvement of cytosine, formerly described as a coordinator of cell metabolism in SARS-CoV-2. Interestingly, tryptophan represents a metabolic node that involves the synthesis of serotonin, the kynurenine pathway, and the indole/aryl hydrocarbon receptor (AHR) pathway. Indole acetic acid is a ligand of AHR that has been linked to many disorders involving immune and inflammatory processes [64]. Moreover, the evaluation of serum from 33 patients with COVID-19 in comparison with 19 negative controls showed an altered tryptophan metabolism in the kynurenine pathway which seemed to regulate inflammation and immunity; these alterations in tryptophan metabolism correlated with the levels of interleukin-6 (IL-6) [65]. A recent review on SSRIs suggests that they can have a neuroprotective effect during COVID-19 infection. The excessive synthesis of inflammatory mediators may be attenuated by augmenting central and systemic 5-HT levels. SSRIs can blunt the exacerbated immune response in COVID-19 and ameliorate clinical consequences [66].

Some antidepressants, including paroxetine, fluoxetine, mirtazapine, mianserin, desvenlafaxine, venlafaxine, imipramine, amitriptyline, and agomelatine, have effects on the NLRP3–inflammasome complex both in in vitro THP-1 cells stimulated with ATP and in animal models of stress-induced depression or depressed patients. All nine drugs produced an important reduction (between 50% and 60%) of inflammasome activation by the inhibition of IL-1β and IL-18. Furthermore, the serum results were accompanied by a significant decrement in NLRP3 mRNA expression [67].

Some selective serotonin re-uptake inhibitors have been shown to have an effect on endolysosomal trafficking, a characteristic of chemical compounds often called lysosomotropic agents, which has been suggested as a potential therapy for COVID-19 [68].

An additional potential mechanism for immune modulation is sigma-1 receptor agonism, which some antidepressants, including fluvoxamine, have demonstrated [59].

The SSRI fluoxetine is also a functional inhibitor of sphingomyelinase (FIASMAs) that has been shown to have antiviral activity through impairing endolysosomal acidification and cholesterol accumulation in the endosomes (Schloer et al., 2020). Thus, lysosome-targeting drugs have been suggested as a potential treatment for COVID-19 based on preclinical results [68].

In summary, antidepressants may exert anti-inflammatory activity by reducing the levels of proinflammatory cytokines, including interleukin 6 (IL-6), IL-10, tumour necrosis factor (TNF)-alpha, and chemokine (C-C motif) ligand (CCL)-2, or by activating the sigma-1 receptor–IRE1 pathway. Although the detailed specificity of their mechanisms of action against SARS-CoV-2 infection still remains unknown, there is evidence suggesting that antidepressants increase serotonin and may exert their anti-inflammatory effects through cAMP-mediated pathways. Serotonin increases intracellular cAMP levels via serotonin receptors linked to the G protein-mediated stimulation of adenylyl cyclase, leading to a reduction in the expression of cytokines via the inhibition of the protein kinase A (PKA) pathway [56].

## 3. Clinical Evidence

It is well known that the use of antidepressants can have positive effects through alterations related to the neuropsychiatric spectrum associated with the COVID-19 pandemic [69,70], including delirium [71], insomnia [72], suicide [73], and psychosis [73,74]. The existing evidence suggests that a range of 0.9% to 4% of infected people develop psychotic spectrum disorders. 

When it comes to clinical evidence regarding the possible therapeutic efficacy of antidepressants against SARS-CoV-2 infection, several observational studies and two controlled clinical trials have been described in the literature. In addition, there are seven ongoing registered clinical trials that should provide further evidence. The main characteristics of the registered clinical trials are shown in Table 1. 

### 3.1. Observational Studies: SARS-CoV-2 Infection and Mental Health

The COVID-19 pandemic was accompanied by an excess of all-cause mortality at a national level in most countries. There are several factors associated with an increased risk of mortality. The number of published studies addressing the influence of COVID-19 infection in patients receiving mental health services is low. Mental disorders might be a risk factor for severe COVID-19.

A systematic review and meta-analysis was conducted in order to evaluate the risks of COVID-19-related mortality, hospitalization, and intensive care unit (ICU) admission in patients with pre-existing mental disorders and exposure to different psychopharmacological drug classes [75]. The systematic review included 33 studies, and the meta-analysis included 23 studies. It comprised 1,469,731 patients with COVID-19, of whom 43,938 had mental disorders. The results showed an increased risk of COVID-19 mortality (OR 2.00, 95% CI 1.58–2.54) and hospitalization (OR 2.23, 95% CI 1.70–2.94), but no increased risk of ICU admission for the presence of any mental disorder. COVID-19 mortality was found to be associated with exposure to antipsychotics (OR 3.71, 95% CI 1.74–7.91), anxiolytics (OR 2.58, 95% CI 1.22–5.44), and antidepressants (OR 2.23, 95% CI 1.06–4.71). Although a non-protective effect of antidepressants was found, the authors stated that this could have been confounded by the psychiatric indication.

One Danish study evaluated the characteristics and predictors of inpatient hospitalization versus community management and death versus survival, adjusted for sex, age, and comorbidities [76]. It included a Danish nationwide population-based cohort of 228,677 consecutive individuals tested for severe acute respiratory syndrome coronavirus 2 (SARS-CoV-2) RNA from the first COVID-19 case identified on 27 February 2020 until 30 April 2020. A total of 9519 SARS-CoV-2 PCR-positive cases were found, of which 80% were community-managed, 20% were hospitalized (3.2% in an intensive care unit), and 5.5% died within 30 days. Age was found to be a strong predictor of fatal disease, with an odds ratio (OR) from 15 to 90 for 70 to 79 year olds and for subjects older than 90 years when compared to 50 to 59 year olds and adjusted for sex and number of comorbidities. The number of comorbidities was found to be strongly associated with fatal disease (OR 5.2, for subjects with ≥4 comorbidities versus no comorbidities). A total of 82% of lethal cases had at least two comorbidities. A broad spectrum of major chronic diseases were found to be associated with hospitalization and increased mortality, with ORs ranging from 1.2 to 1.3 (e.g., hypertension and ischaemic heart disease) to 2.4 to 2.7 (e.g., organ transplantation and major psychiatric disorder). For mental disorders, the risk of hospitalization or death (adjusted by age/sex) for alcohol abuse was 1.7 (95% CI 1.3–2.3) and 1.8 (95% CI 1.2–2.7), respectively. For substance abuse, the mentioned risks were 1.3 (95% CI 0.9–1.9) and 1.8 (95% CI 1.1–3.2). For major psychiatric disorders, they were 2.1 (95% CI 1.2–3.8) and 2.4 (95% CI 1.1–5.0). In the case of psychopharmacological substances, for the use of benzodiazepines and derivates, the risks were 1.7 (95% CI 1.4–2.1) and 2.0 (95% CI 1.6–2.6), while for antipsychotic use the values were 1.5 (95% CI 1.1–1.9) and 3.3 (95% CI 2.3–4.8). For antidepressant use, the risks for hospitalization and death were 1.3 (95% CI 1.1–1.5) and 1.7 (95% CI 1.3–2.1), respectively. The same research group compared the potential usefulness of psychotropics that functionally inhibited the acid sphingomyelinase/ceramide system (FIASMA) in patients with psychiatric disorders hospitalized for severe COVID-19. The study was observational and retrospective and was carried out at Greater Paris University hospitals. A total of 545 adult inpatients were included and 164 (30.1%) received FIASMA medications at baseline. Its use was found to be significantly associated with a reduced risk of intubation or death (HR = 0.42; 95% CI = 0.31–0.57). The association was not specific to one FIASMA psychotropic class or medication. Those taking FIASMA antidepressants at baseline had a reduced risk of intubation or death compared with those taking a non-FIASMA antidepressants at baseline (HR = 0.57; 95% CI = 0.38–0.86) [77].

A recent observational study conducted in patients hospitalized with COVID-19 evaluated the association of antidepressant use with the risk of intubation or death, comparing patients who received antidepressants and those who did not [78]. The primary outcome was a composite variable including intubation or death in time-to-event analyses. Antidepressant exposure was defined as taking any antidepressant at any time during the follow-up study period from baseline (date of hospital admission) to the end of the hospitalization or intubation or death. This study was conducted in 39 Assistance publique—Hôpitaux de Paris (AP-HP) hospitals. From a total sample of 9509 inpatients with a positive COVID-19 RT-PCR test, a final sample of 7230 adult inpatients were evaluated. A total of 345 patients (4.8%) received an antidepressant within 48 h of their hospitalization. The primary endpoint was achieved in 168 patients exposed to antidepressants (48.7%) and in 1188 subjects who were not (17.3%). After adjusting for factors such as older age and the higher medical severity of patients taking antidepressants, the primary analyses showed a significant negative association of the composite endpoint with exposure to any antidepressant drug (HR, 0.56; *p* < 0.001), SSRI (HR, 0.51; *p* < 0.001), or SNRI (HR, 0.65; *p* = 0.034), but not with other classes of antidepressants. Exposure to fluoxetine, venlafaxine, mirtazapine, and escitalopram was found to be significantly associated with a lower risk of intubation or death (all *p* < 0.05). This association was found at a mean dosage of 21.6 (SD = 14.1) fluoxetine-equivalent milligrams. The conclusion was that treatment with SSRIs and SNRIs could be associated with a reduced risk of death or intubation in COVID-19 patients.

In an observational retrospective study collecting the data of 7995 adult patients with SARS-CoV-2 from the Yale New Haven Health (YNHH) clinical data repository, the influence of baseline factors in relation to admission and in-hospital mortality for patients with SARS-CoV-2 infection as determined by RT-PCR testing was evaluated [79]. Older age was found to be significantly associated with risk of admission, and age ≥85 years was found to lead to the highest risk of admission in multivariable analysis. Male sex was also associated with an augmented risk of admission. The different comorbidities associated with increased risk of admission included obesity (OR 1.18, 95% CI 1.02–1.37), renal failure (OR 1.38, 95% CI 1.08–1.75), drug abuse (OR 1.46, 95% CI 1.11–1.92), peptic ulcer disease (OR 1.47, 95% CI 1.04–2.07), pulmonary circulation disorders (OR 1.53, 95% CI 1.14–2.06), metastatic cancer (OR 1.55, 95% C 1.11–2.15), psychoses (OR 1.98, 95% CI 1.47–2.69), and fluid and electrolyte disorders (OR 1.99, 95% CI 1.67–2.37). Depression was not associated with an increased risk of admission and/or mortality. No data regarding antidepressant use are provided in this manuscript.

A cohort study, using electronic health records from five eastern Massachusetts hospitals from between 1 July 2019 and 4 July 2020, was performed to investigate commonly prescribed medicines that may be associated with a lesser risk of morbidity from COVID-19 [80]. Among 12,818 individuals with COVID-19 testing results available, 2271 (17.7%) tested positive, and 707/2271 (31.1%) were hospitalized. Medications used among individuals who tested positive and who did not require hospitalization included naproxen, ibuprofen, and valacyclovir. Among individuals who had been hospitalized, naproxen and ibuprofen were more commonly prescribed among individuals who did not require ventilation. For antidepressants, the risk for admission and ventilation was not found to be significant in the case of sertraline (OR 0.83, 95% CI 0.35–1.85; OR 0.26, 95% CI 0.01–1.47) and bupropion (OR 0.85, 95% CI 0.34–1.99; OR 3.62, 95% CI 0.96–13.63).

Zimering et al. analysed the baseline risk factors associated with respiratory failure or death in 55 older-adult US military veterans hospitalized for COVID-19 infection during the peak of the pandemic in New Jersey (March–June 2020) [81]. The results found that the non-use (vs. use) of psychotropic medications with serotonin 2A receptor antagonist properties was a significant predictor of an increased risk of death (odds ratio 5.06, 95% CI 1.18–21.7). The serotonin 2A receptor antagonist included mirtazapine and trazodone and second-generation atypical antipsychotics. 

Clelland et al. evaluated whether chronic antidepressant use modifies the risk of COVID-19 infection [82]. This was a retrospective cohort study including chronic in-patient patients in a psychiatric facility operated by the New York State Office of Mental Health. Data collected from electronic medical records included 165 patients observed during the period of January to July 2020. The results showed a significant protective association between antidepressant use and COVID-19 infection (OR 0.33, 95% CI 0.15–0.70). Individual antidepressant classes showed that serotonin–norepinephrine reuptake inhibitors and serotonin-2 antagonist reuptake inhibitors presented this protective effect. 

In a recent observational retrospective study, Rauchman et al. evaluated whether patients already on SSRIs upon hospital admission had reduced mortality compared to patients not on chronic SSRI treatment [83]. Data from the electronic medical records of 9044 patients with laboratory-confirmed COVID-19 from six hospitals in the US were collected. The results did not show a significant difference in the risk of dying between patients on chronic SSRIs vs. those not taking SSRIs (adjusted OR 0.96, 95% CI 0.79–1.16). 

A matched case–control study investigated the relationship of prior drug prescription with severe COVID-19 [84]. The study included 4272 cases of severe COVID-19 in Scotland dating from the beginning of the pandemic, and 36,948 matched controls for age, sex, and primary care practice. The number of non-cardiovascular drug classes dispensed was strongly associated with severe COVID-19. The use of tricyclic and related antidepressant drugs in the most recent 120 days was not found to be associated with severe COVID-19 (RR 1.03, 95% CI 0.69–1.54).

In a preliminary observational prospective study of severe COVID-19 infection, lithium carbonate was added to the usual treatment for six patients who also presented psychiatric symptoms which would justify, from a psychiatric point of view, the use of the compound. The results obtained were compared with those of three concomitant control patients not taking lithium. All patients that received lithium carbonate presented improvements in their CRP levels. Similarly, the number of lymphocytes was improved until they returned to normal levels. All six patients on lithium survived, but one of the three controls died. The study presented important limitations, such as the small number of patients and the lack of randomization [85].

Finally, it is interesting to describe the results of a prospective cohort study administering fluvoxamine [86] based on the clinical trial results for fluvoxamine published by Lenze et al. (2020) [87]. In this observational study, a group of home-isolated ambulatory COVID-19 patients were offered the option to receive fluvoxamine and be followed up over the course of 14 days. A total of 65 persons opted to receive fluvoxamine (50 mg twice daily) and 48 declined. The incidence of hospitalization was 0 out of 65 (0%) with fluvoxamine and 6 out of 48 (12.5%) with observation alone. No residual symptoms persisted in the fluvoxamine group, but they did in 60% (29 of 48) under observation only. With the relevant limitations of the design of the study, the results are in line with the published clinical trials regarding available data about the possible protective action of antidepressants in the development of COVID-19.

**Table 1 jcm-11-04038-t001:** Study characteristics of studies registered in CT databases using drugs with antidepressant activity in the treatment of COVID-19 infection.

ID Study Status	Country	Title	Study Phase	Design	No. Patients	Treatment Conditions	Inclusion Criteria	Main Exclusion Criteria	Primary Endpoint
NCT04454307 Not yet recruiting	Egypt	Safety and Efficacy of Tramadol Adjuvant Therapy to Standard Care for COVID-19 Egyptian Patients: A Randomized Double Blind Controlled Clinical Trial	II/III	Randomized, double-blind, controlled	100	Tramadol 100 mg twice daily for 10 days Standard care + placebo	Patients 18–65 years old (both sexes) Newly diagnosed symptomatic COVID-19 patients with mild-to-moderate respiratory manifestations	Patients with abnormal liver function, chronic kidney disease or dialysis Immunocompromised patients Subjects with a comorbid condition such as hypertension, cardiovascular disease, diabetes mellitus, asthma, COPD, malignancy	Number of COVID-19 PCR (10 days)
NCT04342663 Completed (Lenze et al. [87])	US	A Double-blind, Placebo-controlled Clinical Trial of Fluvoxamine for Symptomatic Individuals with COVID-19 Infection (Stop COVID-1)	II	Randomized, double-blind, placebo-controlled	152	Fluvoxamine 100 mg capsules, three times daily for 15 days Placebo	Patients 18 and older (both sexes) Not hospitalized Recently tested SARS-CoV-2 (COVID-19 virus)-positive Currently symptomatic with one or more of the following: fever, cough, myalgia, mild dyspnoea, diarrhoea, vomiting, anosmia, ageusia, or sore throat	Illness severe enough to require hospitalization or already meeting the study’s primary endpoint for clinical worsening Unstable medical comorbidities such as severe underlying lung disease, decompensated cirrhosis, congestive heart failure Immunocompromised patients	Clinical deterioration (15 days) (follow up: 30 days)
NCT04668950 Completed	US	Fluvoxamine for Early Treatment of COVID-19: a Fully remote, Randomized Placebo Controlled Trial (Stop COVID-2)	III	Randomized, double-blind, placebo-controlled	1100 (planned) 683 (when enrolment was stopped)	Fluvoxamine Up to 200 mg per day (2 capsules) as tolerated, for approximately 15 days Placebo	Patients 18 and older (both sexes) Not hospitalized Recently tested SARS-CoV-2 (COVID-19 virus) positive Currently symptomatic with one or more of the following: fever, cough, myalgia, mild dyspnoea, chest pain, diarrhoea, nausea, vomiting, anosmia, ageusia, sore throat, nasal congestion	Illness severe enough to require hospitalization or already meeting the study’s primary endpoint for clinical worsening Unstable medical comorbidities Immunocompromised patients Taking drugs that can interact with fluvoxamine Received vaccine for COVID-19	Clinical deterioration (15 days) (follow up: 30 days)
NCT04377308 Recruiting	US	Fluoxetine to Reduce Intubation and Death After COVID19 Infection	IV	Non-randomized, open, controlled (patients may choose to take fluoxetine or usual treatment)	2000	Fluoxetine 20 mg to 60 mg daily (2 weeks–2 months) duration No intervention: treatment as usual	Patients aged 18 and over COVID-19 test positive or presumptive positive awaiting COVID-19 testing or results by the following criteria: fever, cough and shortness of breath, or presumptive positive by one of these three criteria (fever, cough or shortness of breath) and known exposure to COVID-19 positive individual in past 2 weeks	According to SmPC for fluoxetine	Hospitalizations Intubation Death (2 months)
NCT04510194 Recruiting	US	COVID-19-OUT: Outpatient Treatment for SARS-CoV-2 Infection, a Factorial Randomized Clinical Trial COVID-OUT	II/III	Randomized, quadruple-masked, placebo-controlled	1160	Metformin 1500 mg daily for 14 days Ivermectin (390 mcg/kg if weight <104 kg and 470 mcg/kf if weight >104 kg) for 3 days Fluvoxamine 50 mg bid for 14 days Metformin + Fluvoxamine Metformin + Ivermectin Placebo	Positive laboratory test for active SARS-CoV-2 viral infection (i.e., +PCR) within 3 days of randomization GFR > 45 mL/min within 2 weeks for patients >75 years old, or with history of heart, kidney, or liver failure	Hospitalized for COVID-19 or other reasons Symptom onset greater than 7 days before randomization Immune compromised state, hepatic impairment, severe kidney disease, unstable heart failure	Decreased oxygenation Emergency Department utilization (14 days) Post-Acute Sequelae of SARS-CoV-2 Infection (PASC) Questionnaire (6 and 12 months)
NCT04718480 Recruiting	Hungary	A Randomized, Double-blind, Placebo-controlled, Adaptive-design Study to Assess the Safety and Efficacy of Daily 200 mg Fluvoxamine as add-on Therapy to Standard of Care in Moderate Severity COVID-19 Patients	II	Randomized, double-blind, placebo-controlled, adaptive design add-on treatment	100	Fluvoxamine (2 × 100 mg) daily (with careful dose escalation and tapered dose reduction) for 74 days Placebo	Patients aged 18–70 years old (both sexes) Hospitalized patients with confirmed SARS-CoV-2 by PCR or known contact of confirmed case with consistent symptoms Moderate cases	Mild COVID-19 at randomization Severe COVID-19 at randomization Critical COVID-19 at randomization	Time to clinical recovery after treatment (74 days)
NCT04885530 Recruiting	US	ACTIV-6: COVID-19 Outpatient Randomized Trial to Evaluate Efficacy of Repurposed Medications	III	Randomized, double-blind and placebo controlled within each treatment arm (allocation to the specific study drug is known by patient and study team)	15,000	Ivermectin 7 mg daily for 3 days or placebo Fluvoxamine 50 mg bid for 10 days or placebo Fluticasone 200 mg powder for inhalation od for 14 days or placebo	Age ≥30 years old Confirmed SARS-CoV-2 infection by PCR or antigen test within 10 days of screening Two or more current symptoms of acute infection for ≤7 days	Prior diagnosis of COVID-19 infection (>10 days from screening) Current or recent (within 10 days of screening) hospitalization	Number of hospitalizations Number of deaths Number of symptoms (14 days)
NCT04727424 Recruiting Published fluvoxamine vs. placebo results (Reis et al. [88])	Brazil	A Multicenter, Prospective, Adaptive, Double-blind, Randomized, Placebo-controlled Study to Evaluate the Effect of Fluvoxamine, Ivermectin, Doxasozin and Interferon Lambda 1A in Mild COVID-19 and High Risk of Complications	III	Randomized, quadruple-masked, placebo-controlled	3645	Fluvoxamine 100 mg daily for 9 days Doxazosin 2 mg, up to 8 mg/day for 3 days Ivermectin 6 mg od for 3 days Peginterferon Lambda-1a single dose of 180 mcg single SC Peginterferon Beta-1A prefilled syringe (single dose of 125 mcg CS) Placebo	Patients aged 18 or older Patient with positive rapid test for SARS-CoV2 antigen performed on the screening or with a positive SARS-CoV2 diagnostic test within 7 days of the onset of symptom Patients with at least one enhancement criteria for disease complication	Negative SARS-CoV2 test Patients with COVID-19 being referred for hospitalization Patients with clinical evidence of moderate disease and/or hospitalization indication	Rate of active comparators in changing the need for emergency care AND observation for more than 6 h Rate of active comparators in changing the need for hospitalization
NCT05087381 Recruiting	Thailand	Randomized-controlled Trial of the Effectiveness of COVID-19 Early Treatment in Community with Fluvoxamine, Bromhexine, Cyproheptadine, and Niclosamide in Decreasing Recovery Time	IV	Randomized, open-label, multiarm, prospective, adaptive platform, controlled	1800	Fluvoxamine 150 mg daily (50 mg in the morning and 100 mg at bedtime) for 14 days. Fluvoxamine + bromhexine 16 mg daily for 10 days Fluvoxamine + Cyproheptadine 12 mg daily for 14 days Niclosamide 1000 mg daily for 14 days Niclosamide + Bromhexidine Standard of care (according to Thailand ministry guidelines)	Patients aged 18 or older Patients with mild symptoms consistent with COVID-19 confirmed by antigen test or PCR	Severe hepatic impairment Severe renal impairment	Hospital admission or mortality related to COVID-19 Time taken to self-report recovery Progression to severe COVID-19 disease (28 days)

Table content extracted from clinicaltrials.gov. Only ongoing and completed studies have been included.

### 3.2. Published Clinical Trials Assessing the Efficacy of Drugs with Antidepressant Activity in the Treatment of COVID-19

In order to confirm the possible efficacy of antidepressant drugs against SARS-CoV-2 infection, further randomized clinical trials are required. In this sense, two studies testing fluvoxamine have recently been published [87,88]. A summary of the registered information can be found in Table 1.

In the study conducted by Lenze (STOP COVID 1), a total of 152 non-hospitalized patients were randomly assigned to groups administered 100 mg of fluvoxamine or placebo three times daily for 15 days, with a 30-day follow up. The study results showed that the administration of fluvoxamine at the early stages of the disease significantly reduced the clinical deterioration, showing a good safety profile. Thus, after 15 days of treatment, 0 out of 80 patients in the fluvoxamine group and 6 out of 72 (8.3%) in the placebo group presented clinical deterioration (absolute difference, 8.7% [95% CI, 1.8–16.4%] from survival analysis; log-rank *p* = 0.009). In view of the positive results of this study, the same investigation group conducted a phase III trial with a very similar design and a considerably larger sample size (1100 planned patients) in order to confirm the results of the phase II study (STOP COVID 2) [89]. After an interim analysis, the Data Safety Monitoring Board advised the early termination of the study for futility due to a low case rate and recruitment difficulties [90].

The second published clinical trial testing the use of fluvoxamine as a potential treatment for COVID-19 consisted of a placebo-controlled adaptive platform trial (the TOGETHER STUDY) [88], where several medications were tested in confirmed SARS-CoV-2 patients with a known risk factor for progression to severe disease in order to evaluate the reduction in the need for hospitalization, defined as either the need for emergency care observation longer than 6 h or hospitalization due to lower tract respiratory infection. In the arm testing fluvoxamine, a total of 1497 patients were randomly assigned to receive 100 mg of this medication twice a day for 10 days (*n* = 741) or a matched placebo (*n* = 756). The study results showed a lower proportion of patients with an emergency setting observation period >6 h or hospitalization due to COVID-19 in the fluvoxamine group compared to placebo (79 (11%) vs. 119 (16%); relative risk = 0.68; 95% Bayesian credible interval [95% BCI]: 0.52–0.88). The fluvoxamine arm was stopped for superiority, since the prespecified superiority threshold was surpassed (99.8% vs. the prespecified 97.6% (risk difference 5.0%)). The number of deaths was lower in the fluvoxamine group than in the placebo group: 17 vs. 25 (odds ratio = 0.68, 95% CI: 0.36–1.27). There were no significant differences in safety results for either treatment condition.

### 3.3. Ongoing Clinical Trials Assessing the Efficacy of Drugs with Antidepressant Activity in the Treatment of COVID-19

Further possible evidence of the effect of some antidepressants on the treatment of COVID-19 is currently being tested in seven ongoing clinical trials [91,92,93,94,95,96,97]. Five of these are testing fluvoxamine, one is testing fluoxetine, and one is testing tramadol which, although pharmacologically classified as analgesic, has a dual mechanism of action: inhibiting serotonin and norepinephrine reuptake and binding to μ-opioid receptors. The rationale of the investigators testing these drugs as possible treatments for COVID-19 is based on the possible prevention of cytokine storm through the influence of the S1R–IRE1 pathway in the case of fluvoxamine [89], the inhibition of IL-6 for fluoxetine [92], and the anti-inflammatory effect decreasing the plasma level of TNF-α in the case of tramadol [91]. The main characteristics of these studies are described in Table 1.

## 4. Discussion

Our results show some evidence from preclinical data and observational studies regarding the possible efficacy of some specific antidepressants for COVID-19 infection. In addition, we found two already published clinical trials showing positive results after the administration of fluvoxamine versus a placebo, in terms of clinical deterioration and the rate of observation unit state or hospitalization, respectively [87,88]. Evidence from clinical trials testing tramadol and fluoxetine as potential treatments for COVID-19 is not shown, since the clinical trials found are ongoing.

As a result of the high level of interest in the clinical trials conducted on fluvoxamine, two meta-analysis have recently been published [98,99]. The studies included in both cases were the STOP COVID 1 and 2 and the TOGETHER studies. The conclusions in both cases were that fluvoxamine showed a high probability of being associated with reducing the rate of hospitalization in outpatients with COVID-19 [98], and either reducing clinical deterioration or hospitalizations [99].

Apart from the evidence found in the clinical trials, the possible positive activity found regarding COVID-19 infections is also based on two observational studies. In one, a reduction in the composite variable of intubation or death was observed with the previous use of any antidepressant and the use of SSRIs and SNRIs. When taking SSRIs such as escitalopram and fluoxetine, or an SNRI such as venlafaxine, in all cases, a significantly lower risk of intubation or death was observed [78]. In the other study, a reduced risk of intubation or death was observed in those taking antidepressants [77]. The use of antidepressants was associated with positive but not significant results. The risk of admission and ventilation for sertraline showed an OR of 0.83 and an OR of 0.26, respectively, but in both cases the results were non-significant [80]. Results from another study showed a significant protective effect of antidepressant use for COVID-19 infection (OR of 0.33) [82].

However, in another observational population study [76], the results were negative, and antidepressant use was found to be associated with an increased risk of hospital admission (OR 1.3) and death (OR 1.7), though both non-significant. Similar results were obtained in a meta-analysis on antidepressant use, finding it to be associated with a major risk of death (OR 2.23) but not with hospitalization or ICU admission [75]. In a third study, the results did not show any difference in the risk of dying between patients on chronic SSRIs and controls [83].

The possible negative effects of treatment with antidepressants should also be considered, such as possible drug–drug interactions with standard COVID-19 treatments. The combination of hydroxychloroquine or chloroquine, which are used in the early stages, with some antidepressants such as citalopram, escitalopram, venlafaxine, or clomipramine may increase the risk of QT prolongation and lead to the development of potentially dangerous cardiac arrhythmias (torsade de pointes). These drugs should not be coadministered [100].

The main limitations of this narrative review are the use of only the MEDLINE (PubMed) and MedRxiv/BioRxiv databases for published articles and some of the public clinical trials registries, and the inclusion of only publications in English. We included one preprint article, which is a report that has not been peer-reviewed. We do not know if this report will end up being published in a standard journal. However, at the time of the pandemic, sharing data is considered to be an adequate way to disseminate knowledge in a short period of time in order to help basic and clinical researchers make decisions regarding investigations and managing patients. Another limitation of this study is the small number of published studies available, the fact that most of them are observational and retrospective, and the scarce final results available from registered clinical trials, with only two studies having been published.

## 5. Conclusions

In conclusion, there is some basic research that suggests a possible benefit of some antidepressants based on the antiviral activity of several compounds and the anti-inflammatory effect of these substances. The clinical evidence is scarce, but it seems that antidepressant use—specifically, the use of fluvoxamine—can reduce time to clinical deterioration, the time of emergency setting observation and hospitalization, and the need for invasive ventilation/intubation in some patients. The sum of this antiviral and anti-inflammatory research and the results from a couple of observational studies and the two clinical trials available provide the basis for the remaining ongoing clinical trials and possible further ones evaluating the experimental efficacy of the use of antidepressants for COVID-19 infection in humans. As in other cases of repurposing therapies, until stronger evidence from the ongoing clinical trials is published, the current evidence does not recommend the use of antidepressants for the treatment of COVID-19 infections, except in subjects meeting selection criteria in a clinical trial.

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
