# Peer review of "Antidepressant Drugs and COVID-19: A Review of Basic and Clinical Evidence"

_jcm, 2022, doi:10.3390/jcm11144038_

Round 1

Reviewer 1 Report

1. plz avaoid using pronouns/prepositions in the begining of senetences.

for eg. There is evidence to support the theory that some antidepressants can reduce the concentrations of different cytokines in humans and animals and, recently, an antiviral activity of some antidepressants against SARS-CoV-2 has been reported. Although the evidence is limited, the sum of the antiviral and anti-inflammatory preclinical studies and the result.........

2. BioRvix? or https://www.biorxiv.org/

3. Avoid repetition of wrods in paragraphs

4. Introduction, first three paragraphs were mentioned in lot of the review papers. For preseneting novel review avoid these paragraphs and simplify in 3-4 lines.

5. in the introduction, most of the paragraphs are repetitive from other published reports. avoid these repeats.

6. Plz shorten 1.1.2. Inflammatory phase

7.  sections "Clinical trials assessing the efficacy of drugs with antidepressant activity in the treatment of 425 COVID-19" is not written well based on the given concept

8. Add the Table for section "3. Clinical evidence"

9. Paper has lot grammar and syntax errors.

Reviewer 2 Report

It would be helpful to understand if you provide any mechanism or how antidepressants act to reduce the inflammation. The second point is there is any effects of this antidepressants on the health, if possible provide the data on it  from the published literature for better  understanding.

Is there any data available between COVID-19 and without COVID-19 patient taking the antidepressants ?
